



Impact of Siberia forest fires on the atmospheric environment over the
Korean Peninsula during summer 2014
Jinsang Jung [a,*], Youngsook Lyu[b], Minhee Lee[b], Taekyung Hwang[b], Youdeok Hong[b],
Jihyung Hong[b], Sangil Lee[a], Sanghyub Oh[a]
[a]Center for Gas Analysis, Korea Research Institute of Standards and Science (KRISS),
Daejeon 34113, Republic of Korea
[b]Department of Climate and Air Quality Research, National Institute of Environmental
Research, Daejeon 34944, Republic of Korea
Running title: LRT Russian FF
Last modified: January 19, 2015
Submitted to Atmospheric Chemistry and Physics
*Corresponding author: Jinsang Jung (jsjung@kriss.re.kr)





**Abstract**
Extensive forest fires occurred during the late July, 2014 across the Siberia forest
region, Russia. Smoke plumes emitted from the Siberia forest fires were long-range
transported through Mongolia and northeast China, and down to the Korean Peninsula,
which is located at ~3,000 km south of the Siberia forest. Notably high aerosol optical
depth (AOD) of ~4 was observed at a wavelength of 500 nm near the source region of
the Siberia forest fires. The smoke plumes reached about 3–5 km height near the source
region and then below 2 km height near the Korean Peninsula. Elevated concentration
of levoglucosan was observed as an average of $119.7 \pm 6.0$ ng m$^{-3}$ (mean ± one standard
deviation) which was ~4.5 times higher than those observed during the non-event period
in July, 2014. During the middle of July 2014, another type of haze episode occurred
that was mainly caused by long-range transported haze plumes originated from urban
and industrial complexes in the East China. Sharp increases in $SO_4^{2-}$ concentrations
$(23.1 \pm 2.1$ µg m$^{-3})$ were observed during the Chinese haze episode. The haze episode
caused by the long-range transported Siberia forest fires was clearly distinguished with
relatively high OC/EC ratio $(7.18 \pm 0.2)$ and OC/$SO_4^{2-}$ ratio $(1.31 \pm 0.07)$ compared to
those (OC/EC ratio: $2.4 \pm 0.4$, OC/$SO_4^{2-}$ ratio: $0.21 \pm 0.05$) during the Chinese haze
episode. Remote measurement techniques and chemical analyses of the haze plumes
clearly showed that the haze episode occurred during the late July, 2014 was mainly
caused by the long-range transported smoke plumes emitted from Siberia forest fires.





## 1. Introduction

Forest fires emit large amounts of gaseous and particulate pollutants into the atmosphere, namely carbon dioxide ($CO_2$), carbon monoxide (CO), methane ($CH_4$), nitrogen oxides ($NO_x$), ammonia ($NH_3$), particulate matter (PM), non-methane hydrocarbon (NMHC), and other chemical species (Crutzen and Andreae, 1990). These gaseous and particulate pollutants can alter regional climate in downwind area by altering ambient temperature, cloud property and the efficiency of precipitation (Jeong et al., 2008; Youn et al., 2011; Jeong et al., 2014). They can also influence the air quality of downwind areas in urban, ocean, and Arctic regions through long-range atmospheric transport (Carvalho et al., 2011; Quennehe et al., 2012; Schreier et al., 2015).

During the severe forest fires smoke episode in Moscow, Russia in August 2010, notably high concentrations of total carbon (average 202 µg m$^{-3}$) and levoglucosan (3.1 µg m$^{-3}$) were observed with elevated organic carbon/elemental carbon (OC/EC) ratio of 27.4 (Popovicheva et al., 2014). Total carbon concentration during the severe smoke episode exceeded 10 times that during the non-event period in Moscow, Russia (Popovicheva et al., 2014). During the severe forest fires smoke episode in Siberia region in May 2003, the surface $PM_{10}$ and $O_3$ concentrations in the downwind areas increased by 5−30 µg m$^{-3}$ and 3−20 ppbv, respectively, having an important implication for air quality over East Asia (Jeong et al., 2008).

The territory of Russian Federation is covered with over 800 million hectares of forest, which is equal to 50 billion tons of growing carbon stock, where annually about 1% is damaged by fires (Bondur, 2010; Popovicheva et al., 2014). Russian boreal forests are subject to frequent wildfires. Each year, 10–35 thousand forest fires covering 5,000–53,000 km$^2$ (including 4,000–10,000 km$^2$ of high intensity, stand-replacing fires)





were detected in actively protected portions of the Russian forest (Bartalev et al., 1977;
Isaev et al., 2002; Mei et al., 2011). Siberia is one of the world's major boreal forest fire
areas as approximately 12,000−34,000 wildfires occurred every year in Russia for the
period 1974−1993 (Conard and Eduard, 1996).
Frequent forest fires over the Siberia region had an impact on downwind areas in
Mongolia, China, Korea and Northwestern Pacific through long-range atmospheric
transport (Kajii et al., 2002; Kanaya et al., 2003; Lee et al., 2005; Jeong et al., 2008;
Youn et al., 2011). In May 2003, intense forest fires occurred over Siberia (Lee et al.,
2005; Jeong et al., 2008; Youn et al., 2011). Satellite observation clearly showed the
transport of the smoke plume emitted from the Siberia forest fires through Mongolia
and eastern China, down to the Korean Peninsula (Lee et al., 2005). Simulation results
by Youn et al. (2011) showed a significant surface cooling of −3.5 K over Siberia forest
region. The simulation also showed that smoke aerosols affected the large-scale
circulations and resulted in the increases in rainfall rates of 2.9 mm day$^{-1}$ averaged over
the northern west Pacific. Jeong et al. (2008) reported that the smoke plume from the
Siberian forest fires in May 2003 acts mainly as a cooling agent, resulting in a negative
radiative forcing of −5.8Wm$^{-2}$ at the surface over East Asia.
Severe wildfires occurred in the Russian forest region during summer, 2014.
Intensity of wildfires during the summer, 2014 appears to be three times bigger than
2013. According to NASA MODIS FIRMS (Fire Information for Resource Management
System), daily average of ~5,000 active fires were detected in the Siberia forest region
covering     Irkutsk     to     Yakutsk     area     during     15–25     July     2014
(https://earthdata.nasa.gov/active-fire-data-tab-content-6). MODIS satellite RGB images



clearly showed that these smoke plumes lasted more than a week and transported
southern direction down to Mongolia, northern China and the Korean Peninsula.

In this study, we investigate the smoke plumes emitted from Siberia forest fires

during the late July, 2014 and their long-range atmospheric transport to the Korean
Peninsula. Spatial transport mechanism of the smoke plumes is investigated based on
satellite image analyses and satellite-based lidar observation. We also characterize the
chemical composition of the long-range transported smoke plumes reached at the
Korean Peninsula. In contrast to the forest fire plume event, chemical characteristics of
long-range transported anthropogenic pollutants from East China are also investigated.

2.  Experimental Methods
2.1 Atmospheric aerosol sampling and sample preparation

Daily $PM_{2.5}$ (particulate matter with a diameter less than or equal to 2.5 micrometers)

sampling was carried out at a central region air quality monitoring station (36.19 °N,
127.24 °E) in Daejeon, Korea from 1 to 31 July 2014. $PM_{2.5}$ samples were collected on
pre-baked quartz fiber filters (Pall-Life Sciences, 47 mm diameter) using an aerosol
sampler (APM Korea, model PMS-103) at a flow rate of 16.7 L $min^{-1}$ on the rooftop of
a comprehensive monitoring building (~15 m above the ground) of National Institute of
Environmental Research in Korea. Before and after sampling, the filter samples were
stored in a freezer wrapped with aluminum foil at –20 ℃. A total of 31 filter samples
were collected in this study, and additional field blank filters were collected before and
after the sampling period.

Ultrapure water used in this study was prepared using a Labpure S1 filter and a

ultra-violet (UV) lamp (ELGA, PureLab Ultra). Resistivity and total organic carbon




(TOC) values of the ultrapure water were maintained as 18.2 MΩ cm$^{-1}$ and 4 ppb,
respectively. To measure carbohydrates and water-soluble ions, a quarter of each filter
sample was extracted with 10 mL of ultrapure water under ultrasonication (for 30 min)
and then passed through a disk filter (Millipore, Millex-GV, 0.45 mm). Water extracts
were stored in a refrigerator at 4 °C before analysis.

2.2 Analysis of chemical composition of fine particles
Mass concentration of PM$_{2.5}$ was measured using a beta-attenuation technique (Met
One Instruments, BAM 1020), with an hourly averaging time resolution. The detection
limit and measurement error of the beta-attenuation technique were reported as 3.6 μg
m$^{-3}$ and 8 %, respectively by the manufacturer. In addition to PM$_{2.5}$ mass concentration,
chemical composition of daily PM$_{2.5}$ was also characterized through filter sampling and
laboratory analysis. Because the time interval of chemical composition of PM$_{2.5}$ was
daily basis, daily average PM$_{2.5}$ mass data was calculated from average PM$_{2.5}$ mass data
and used in this study.

2.2.1   Levoglcosan and mannosan analysis
Levoglucosan and mannosan were analyzed by an improved high-performance
anion-exchange chromatography (HPAEC) method with pulsed amperometric detection
(PAD) (Engling et al., 2006; Jung et al., 2014). The HPAEC-PAD system uses an ion
chromatograph consisting of an electrochemical detector and gold electrode unit, along
with an AS40 auto-sampler (Thermo Fisher Scientific, Dionex ICS-15000).
Levoglucosan and mannosan were separated by a CarboPak MA1 analytical column (4
x 250 mm) and a sodium hydroxide solution as an eluent. The detection limit of



levoglucosan and mannosan was 3.0 and 0.7 ng m$^{-3}$, respectively. The analytical error,
defined as the ratio of the standard deviation to the average value, obtained from
triplicate analyses of filter samples, was 1.9% and 0.73%, for levoglucosan and
mannosan, respectively.

2.2.2    Water-soluble inorganic ions analysis
Water-soluble inorganic ions were analyzed using an ion chromatography (Thermo
Fisher Scientific, Dionex ICS-15000). The anions; nitrate ($NO_3^-$) and sulfate ($SO_4^{2-}$),
were separated using an IonPAC AS15 column with an eluent of 20 mM of potassium
hydroxide (KOH) at flow rate of 0.5 mL min$^{-1}$. The detection limits of $NO_3^-$ and $SO_4^{2-}$,
which are defined as 3 times standard deviation of field blanks, were 0.01 and 0.11 μg
m$^{-3}$, respectively. The analytical errors of $NO_3^-$ and $SO_4^{2-}$ were 2.3% and 1.7%,
respectively. The cations, sodium ($Na^+$), ammonium ($NH_4^+$), potassium($K^+$),
calcium($Ca^{2+}$), and magnesium ($Mg^{2+}$), were separated using an IonPac CS-12A column
(4 x 250 mm) with an eluent of 38 mM of methanesulfonic acid (MSA) at a flow rate of
1.0 mL min$^{-1}$. The detection limits of $NH_4^+$ and $K^+$ were 0.03 and 0.006 μg m$^{-3}$,
respectively. The analytical errors of $NH_4^+$ and $K^+$ were 1.4% and 0.73%, respectively.

2.2.3    Organic carbon/elemental carbon analysis
PM$_{2.5}$ carbonaceous aerosol was measured using a semi-continuous organic
carbon/elemental carbon (OC/EC) analyzer (Sunset Lab., Model RT3140). The air
samples were drawn at 8 L min$^{-1}$ through a PM$_{2.5}$ sharp-cut cyclone. The sampled
aerosols then were passed through a multichannel parallel plate denuder with a carbon-
impregnated filter to remove semi-volatile organic vapors, and then collected on a



quartz-fiber filter. The sampled aerosols were analyzed based on thermal-optical
transmittance (TOT) protocol for pyrolysis correction and NIOSH (National Institute
for Occupational Safety and Health) 5040 method temperature profile (Birch and Cary,
1996; Jung et al., 2010). External calibration was performed using known amounts of
sucrose. The detection limit of both OC and EC was 0.5 µg C m$^{-3}$ for 1 hr time
resolution reported by the manufacturer. The uncertainty of OC and EC measurements
is reported as 5% (Polidori et al., 2006).

2.3 Satellite aerosol optical depth and air mass backward trajectories
The NOAA/ARL HYSPLIT (HYbrid Single-Particle Lagrangian Trajectory) air
mass backward trajectory analysis (Draxler and Rolph, 2015; Rolph, 2015) and
Moderate Resolution Imaging Spectro-radiometer (MODIS) satellite image analysis
were used to characterize potential source regions and the transport pathway of the haze
plume. Air mass backward trajectories ended at the measurement site were computed
for 200, 500 and 1000 m above ground level (AGL) heights using the HYSPLIT model.
All back-trajectories were calculated at 00:00 UTC and 12:00 UTC (09:00 LT and
21:00 LT, respectively) extending to 96 h backward with 1 h time interval. The
calculated air mass pathways indicate the general airflow pattern rather than the exact
pathway of air masses because the typical error of the traveled distance are up to 20%
for the trajectories computed from analyzed wind fields (Stohl, 1998),.
Aerosol optical thickness (AOT) data retrieved by the V5.2 version of the NASA
MODIS algorithm, called Collection 005 (C005) (Levy et al., 2007a, b) were used in
this study. AOT data, which is part of the MODIS Terra/Aqua Level-2 gridded
atmospheric   data   product,   are   available   on   the   MODIS   web   site



(http://modis.gsfc.nasa.gov/). Cloud-screened Level 1.5 sunphotometer data at Yakutsk
site (61.66 °N, 129.37 °E, 118 m above sea level) and Ussuriysk site (43.70 °N,
132.16 °E, 280 m above sea level) in Russia were obtained from the AERONET site
(http://aeronet.gsfc.nasa.gov). This studt used total column-integrated spectral aerosol
optical thickness (AOT) determined by the AERONET algorithm (Dubovik and King,

2000).

CALIOP (Cloud-Aerosol Lidar with Orthogonal Polarization) is a space based lidar

system onboard the Cloud Aerosol Lidar and Infrared Pathfinder Satellite Observations
(CALIPSO) satellite launched in 2006 (Winker et al., 2009). This study used version
2.30 data of total attenuated backscatter at 532 nm. Expedited CALIPSO browse images
were        obtained        from        the        CALIPSO        website        (http://www-
calipso.larc.nasa.gov/products/lidar/browse_images/show_calendar.php).

3.   Results and Discussion
3.1    Overview of chemical composition of fine particulate matter(PM$_{2.5}$)

Figure 2 shows temporal variations of chemical compositions of PM$_{2.5}$ at the

Daejeon site during the entire measurement period. Daily average PM$_{2.5}$ mass
concentrations ranged from 8.0 μg m$^{-3}$ to 65.1 μg m$^{-3}$ with an average of 27.5 ± 15.2 μg
m$^{-3}$. Two peaks of PM$_{2.5}$ mass concentration were obtained during 12−16 July (first
episode) and 27−28 July 2014 (second episode). PM$_{2.5}$ mass concentrations reached to
65.1 μg m$^{-3}$ and 56.2 μg m$^{-3}$ during the first and second episodes, respectively. The
temporal variation of the sum of PM$_{2.5}$ chemical compositions showed a similar pattern
with that of total PM$_{2.5}$ mass as shown in Fig. 2. During the entire measurement period,
SO$_4^{2-}$ was found as the highest value with an average of 8.8 ± 7.0 μg m$^{-3}$, followed by





OC (4.3 ± 2.0 µg m$^{-3}$), NH$_4^+$ (4.3 ± 3.3 µg m$^{-3}$), EC (1.1 ± 0.4 µg m$^{-3}$) , and NO$_3^-$ (1.0 ±
1.1 µg m$^{-3}$) with minor contributions from Ca$^{2+}$, K$^+$, and Na$^+$.

3.2    Classification of haze episodes during summer, 2014
3.2.1 Long-range transported smoke plumes from Siberia forest fires

MODIS RGB images clearly show severe smoke plumes over the Siberia forest

region during the late July, 2014. Figure 3a shows a typical example of satellite RGB
images of the smoke plumes emitted from Siberia forest fires and their atmospheric
transport to south during 25 July 2014. Fire events in the Siberia forest region were
from the MODIS active fire product (Gilio et al., 2003) and expressed as red dots in
Fig. 3a. It is clearly seen that the smoke plumes originated from the Siberia forest fires
lingered to the southern direction of the Korean Peninsula across Mongolia and
northeast China. HYSPLIT backward trajectory analyses in Fig. 3b also show that air
masses originated from the Siberia forest region transported to the Korean Peninsula
during 26–28 July 2014.

Figure 4 shows horizontal distribution of aerosol optical depth (AOD) over the East

Asia during 23–28 July 2014. High loading of AOD was clearly shown over the
Siberia forest region on 23 July when forest fires occurred. The transport of high
loading of AOD was clearly seen down to northeast China and further to the Korean
Peninsula from 23 July to 28 July 2014 (Fig. 4). These horizontal distributions of AOD
also support the transport of smoke plumes emitted from the Siberia forest fires into
the Korean Peninsula during the late July, 2014.

Figure 5 shows temporal variations of AODs measured by a sunphotometer at the

Yakutsk and Ussuriysk sites. The Yakuksk site is located near the source region of the



Siberia forest fires whereas the Ussuriysk site is located close to the north of the
Korean Peninsula as shown in Fig. 3. AOD measured at the Yakutsk site started to
increase from 23 July and high AOD continued until 26 July 2014. The maximum
AOD reached at ~4 at the Yakutsk site during 24 July 2014 when the Siberia forest
fires occurred. The high loading of AOD lasted for 4 days at the Yakutsk site during
the Siberia forest fires episodes. Interestingly, a sharp increase in AOD was also
observed at the Ussuriysk site during 24 July 2014. This result implied the rapid
transport of the smoke plumes to the northern Korean Peninsula within one day.
Figure 6 shows MODIS RGB images and vertical distributions of total attenuated
backscatter at a wavelength of 532 nm measured by the CALIPSO satellite during 24,
25, and 27 July 2014. Yellow and red colors in the total attenuated backscatter
measurement in Fig. 6 represent atmospheric aerosols whereas white color represents
cloud. Yellow lines over MODIS RGB images in Fig 6 represent the observation
routes of the CALIPSO satellite. Figure 6a and b clearly showed that smoke layer
existed approximately 3−5 km height near the source region of the Siberia forest fires
during 24 and 25 July 2014. As shown in Fig. 6c, the height of the smoke layer
decreased to below 2 km height during 27 July 2014 when it reached to the Korean
Peninsula.
From the spatial distribution of AOD obtained by MODIS and CALIPSO satellite
observations and HYSPLIT air mass backward trajectory analyses, it was clearly seen
that the smoke plumes originated from the Siberia forest fires during 23−24 July 2014
transported over 3000 km in the southerly direction and had an impact on the Korean
Peninsula during 27−28 July 2014. Ground based AOD measurements by a



sunphometer near the Siberia forest fire area and the Korean Peninsula also supported
the transport of the smoke plume originated from the Siberia forest fires into the
Korean Peninsula. Thus, in this study, the smoke episode during 27−28 July 2014 is
defined as the Siberia forest fire episode.

3.2.2 Long-range transported haze under Asian continental outflow
Besides the haze episode caused by the long-range transported smoke plume emitted
from the Siberia forest fires during the late July, 2014, another haze episode was
observed in the Daejeon site during 14−16 July 2014 as shown in Fig. 2. From the
MODIS RGB image on 14 July in Fig. 7, it was clearly shown that a severe haze plume
originated from East China lingered to the Korean Peninsula across the Yellow Sea.
HYSPLIT backward air mass trajectories also showed transport of air masses originated
from the East China to the Korean Peninsula over the Yellow Sea during 15–16 July

2014.

The East China covering from Beijing to Shanghai region consists of heavily
populated, urbanized, and industrialized cities (Chan and Yao, 2008). Thus, a large
amount of anthropogenic pollutants is emitted from these regions in the East China (Li
et al., in press). Figure 8 shows horizontal distribution of MODIS AOD over East Asia
during 13–16 July 2014. It is clearly seen that the high loading of AOD over East China
lingered to the Korean Peninsula over the Yellow Sea. These results suggest that the
haze episode during 14–16 July 2014 was mainly originated from long-range transport
of pollutants originated from the East China. Thus, in this study, the haze episode
during 14–16 July is defined as the Chinese haze episode.



### 3.3 Chemical characterization of the long-range transported haze plumes

### 3.3.1 Comparison of PM$_{2.5}$ chemical composition during the haze episodes

Figure 9 shows temporal variations of PM$_{2.5}$ mass concentration and its selected chemical components. During the Chinese haze episode, elevated concentrations of SO$_4^{2-}$ (23.1 ± 2.1 µg m$^{-3}$) and K$^+$ (0.27 ± 0.08 µg m$^{-3}$) were obtained whereas elevated concentrations of levoglucosan (119.6 ± 6.0 ng m$^{-3}$), K$^+$ (0.33 ± 0.07 µg m$^{-3}$), and OC (10.8 ± 1.1 µg m$^{-3}$) were measured during the Siberia forest fire episode. As shown in Fig. 9, a similar level of OC was observed during the entire measurement period except the Siberia forest fire episode. However, several peaks of SO$_4^{2-}$ concentrations were observed with the highest peak during the Chinese haze episode.

Figure 10 shows PM$_{2.5}$ mass closures during the Chinese haze and Siberia forest fire episodes. Concentrations of organic aerosols (OM) were reconstructed from measured OC concentrations by multiplying the OM/OC ratio of 1.8 that was measured by an aerosol mass spectrometer in Korea during spring to fall, 2011 under the Asian continental outflow (personal communication from prof. T. Lee). Huang et al. (2011) also reported a similar OM/OC ratio of 1.77 ± 0.08 measured at the downwind site of the highly polluted Pearl River Delta cities in China during the fall, 2008. During the Chinese haze episode, SO$_4^{2-}$ was found as the most dominant species in PM$_{2.5}$ mass with an average contribution of 44.2%, followed by organic aerosols (16.6%) and NH$_4^+$ (19.1%). This result implies that the Chinese haze episode was mainly attributed to anthropogenic pollutants, possibly emissions from industrial complexes and urban cities in the East China. However, during the Siberia forest fire episode, organic aerosol was the most dominant species in PM$_{2.5}$ mass with an average contribution of 38.6%, followed by SO$_4^{2-}$ (16.5%) and NH$_4^+$ (10.0%). The high concentration of



organic aerosols indicated that the Siberia forest fire episode was mainly originated
from biomass burning.

3.3.2 Comparison of biomass burning tracers between two haze episodes in the Daejeon

atmosphere

Levoglucosan, mannosan, and $K^+$ are widely used as an indicator of biomass

burning. Levoglucosan and mannosan are formed during pyrolysis of cellulose and
hemicellulose, and are not emitted from burning of other materials, such as fossil fuels
(Simoneit et al., 1999; Caseiro et al., 2009; Elias et al., 2001). However, caution is
required when $K^+$ is used as a biomass-burning tracer because $K^+$ can also be emitted
from sea salt and soil (Pio et al., 2008). The mass concentrations of biomass burning
tracers and their ratios during the Siberia forest fire and Chinese haze episodes are
summarized in Table 1 and 2.

Significantly elevated concentrations of levoglucosan were observed during the

Siberia forest fire episode whereas relatively much less increases in concentration of
levoglucosan were observed during the Chinese haze episode in Fig. 9. Concentrations
of levoglucosan during the Siberia forest fire episode were measured to be $119.6 \pm 6.0$
ng m$^{-3}$, approximately 6 times higher than those during the Chinese haze episode (22.3
$\pm$ 11.8 ng m$^{-3}$) as shown in Table 1. On the other hand, similar levels of $K^+$ were
obtained during the Chinese haze ($0.27 \pm 0.08$ µg m$^{-3}$) and Siberia forest fire ($0.33 \pm$
$0.07$ µg m$^{-3}$) episodes. Thus, relatively high levoglucosan/$K^+$ ratios were obtained
during the Siberia forest fire episode ($0.37 \pm 0.06$) compared to those ($0.08 \pm 0.03$)
during   the   Chinese   haze   episode   (Table   2).   However,   similar   levels   of
levoglucosan/mannosan ratio were estimated as an average of $3.43 \pm 0.11$ during the





Siberia forest fire episode compared to those during the Chinese haze episodes (4.81 ±
0.41) as shown in Table 2.

Positive correlation was obtained between levoglucosan and OC concentrations

during the Siberia forest fire and Chinese haze episodes in Fig. 11a. OC concentrations
increased simultaneously as $K^+$ concentrations increased during the Siberia forest fire
episode. However, during the Chinese haze episode, relatively small increase in OC
concentrations was observed even though $K^+$ concentrations increased (Fig. 11b). Much
higher OC/EC ratios were obtained during the Siberia forest fire episode (7.18 ± 0.2)
compared to those (2.4 ± 0.4) during the Chinese haze episode (Table 1).

Good correlations of $K^+$ concentration with OC and levoglucosan concentrations

during the Siberia forest fire episode suggest that $K^+$ was mainly originated from the
smoke plume during the Siberia forest fire episode. However, different correlation
patterns between $K^+$ with levoglucosan and OC concentrations were observed during
the Chinese haze episode. This different correlation pattern can be explained as follows.
First, different type of biomass burning might occur during the Chinese haze episode
compared the Siberia forest fire episode. It can be postulated that biomass burning
emissions with relatively lower OC/$K^+$ and levoglucosan/$K^+$ ratio might impact on the
Korean Peninsula during the Chinese haze episode.

Second, $K^+$ during the Chinese haze episode might be originated from other sources

rather than biomass burning. Because OC is predominantly emitted from biomass
burnings, biomass burning particles have relatively high OC/EC ratio and have good
correlation with biomass burning tracers (Cao et al., 2008; Cheng et al., 2008;
Popovicheva et al., 2014). Poor correlations of $K^+$ with OC and levoglucosan
concentrations during the Chinese haze episode suggest that the elevated $K^+$





concentration might be due to emissions from other sources such as soil and sea salt or
industrial complexes. Chow et al. (2008) reported that 3.9%−12.5% of $PM_{2.5}$ consisted
of $K^+$ in stack samples from cement kiln manufacturing process. Positive correlations of
$K^+$ with $SO_4^{2-}$ and EC concentrations in Fig. 9 during the Chinese haze episode also
support that there were additional emission of $K^+$ from anthropogenic sources except
biomass burning.
Elevated concentrations of levoglucosan and OC and relatively high OC/EC ratio
(7.18 ± 0.2) support that the haze episode occurred during the late July, 2014 was
mainly caused by the long-range transported smoke emitted from the Siberia forest fires.
However, significantly elevated $SO_4^{2-}$ concentration with relatively weak increases in
OC and levoglucosan concentrations and relatively lower OC/EC ratio implies that the
Chinese haze episode was mainly caused by anthropogenic pollutants emitted from
industrial complexes and urban cities in the East China with relatively little contribution
of biomass burning.

3.3.3 Tracking major sources of biomass burning during the Siberia forest fire episode
Levoglucosan to mannosan ratios (Levo/Man ratio) and levoglucosan to $K^+$ ratios
(Levo/$K^+$ ratio) during the Siberia forest fire episode are compared with those from
previous chamber and ambient studies in Fig. 12. Hardwood burnings have relatively
higher Levo/Man ratios with a mean value of 28 (range: 2.2−195) (Fine et al., 2001,
2002, 2004a, 2004b; Schauer et al., 2001; Engling et al., 2006; Schmidl et al., 2008a;
Bari et al., 2009; Gonçalves et al., 2010) whereas softwood burning have relatively
lower Levo/Man ratios (mean: 4.3, range: 2.5−6.7) (Fine et al., 2001, 2002, 2004a,



2004b; Schauer et al., 2001; Hays et al., 2002; Engling et al., 2006; Iinuma et al., 2007;
Schmidl et al., 2008a; Gonçalves et al., 2010). Grass (mean: 18, range: 9.2-39) and crop
residue burnings (mean: 29, range: 12−55) have relatively high Levo/Man ratios
compared to leaf burnings (mean: 5.6, range: 5.1−6.0) (Sheesley et al., 2003; ; Engling
et al., 2006, 2009; Sullivan et al., 2008; Schmidl et al., 2008b; Oanh et al., 2011; Cheng
et al., 2013). Levo/Man ratios (mean: 5.3) during the smoke episode in Moscow,
Russian during summer, 2010 are similar to those from softwood and leaf burnings
(Popovicheva et al., 2014).

Because levoglucosan and mannosan are emitted from similar burning processes,

Levo/Man ratio can be used as an indicator to track type of biomass burning. Levo/Man
ratios during the Siberia forest fire episode are similar to those obtained from the
softwood and leaf burning experiments and the smoke episode in Moscow, Russia
during summer, 2010. However, Levo/Man ratios during the Siberia forest fire episode
are much lower than those from the hardwood, grass and crop residue burnings.

Hardwood and softwood burnings have relatively high Levo/$K^+$ ratios, with mean

values of 26 and 46, and ranges of 2.2−195 and 4.6−261, respectively (Fine et al., 2001,
2002, 2004a, 2004b; Schauer et al., 2001; Hays et al., 2002; Engling et al., 2006; Iinuma
et al., 2007; Schmidl et al., 2008a; Bari et al., 2009; Gonçalves et al., 2010). However,
grass, crop residue, and leaf burnings have relatively low Levo/$K^+$ ratios, with mean
values of 3.3, 0.53, and 2.9 and ranges of 0.06−9.5, 0.1−1.2, and 2.4−3.4, respectively
(Sheesley et al., 2003; Engling et al., 2006, 2009; Sullivan et al., 2008; Schmidl et al.,
2008b; Oanh et al., 2011; Cheng et al., 2013). Levo/$K^+$ ratios (mean: 2.8) during the
smoke episode in Moscow, Russian during summer 2010 are similar those from grass,



crop residue, and leaf burnings (Popovicheva et al., 2014).
Levo/K$^+$ ratio during the Siberia forest fire episode is close to those from the grass,
crop residue, leaf burnings and the smoke episode in Moscow but much lower than
those from the hardwood and softwood burnings as shown in Fig. 12b. Levoglucosan
can be removed through photo-oxidative decay during the atmospheric transport
(Hennigan et al., 2010) but K$^+$ is relatively stable in the atmosphere. Laboratory
chamber experiments show that levoglucosan decays as a function of integrated OH
exposure with a typical lifetime of 0.7–2.2 days (Hennigan et al., 2010). Thus, Levo/K$^+$
ratios can decrease during the long-range atmospheric transport. Relatively lower
Levo/K$^+$ ratio during the Siberia forest fire episode was observed compared to those
during the smoke episode in Moscow, Russia during summer, 2010, which can be
explained by photochemical degradation of levoglucosan during the long-range
atmospheric transport.
Based on the comparison of biomass burning tracers from various sources in Fig.
12, it is suggested that smoke aerosols during the Siberia forest fire episode was mainly
originated from the burning of forest leaf in the Siberia region and their long-range
atmospheric transport. Smoke aerosols during the smoke episode in Moscow, Russia
during the summer, 2010 have very similar Levo/Man and Levo/K$^+$ ratios with those
from the leaf burning in Fig. 12. These also support that the smoke episode in the
Russian forest is mainly originated from burnings of forest leaf.

4. Conclusion
This study investigated long-range transported smoke plumes emitted from the
Siberia forest fires occurred during the late July, 2014. Smoke plumes emitted from



Siberia forest fires are generally transported to the Northwest Pacific due to the
prevailing westerlies. However, the haze plume occurred during the late July, 2014 had
a significant impact on the Korean Peninsula located at ~3,000 km south of the Siberia
forest. From spatial distributions of AOD obtained by MODIS satellite, CALIPSO
satellite observation, and HYSPILT air mass backward trajectory analyses, it was
clearly seen that the smoke plumes originated from the Siberia forest fires during 23−24
July 2014 transported over 3,000 km to south direction and had an impact on the
Korean Peninsula during 27−28 July 2014. During that episodic period, elevated
concentrations of levoglucosan (119.6 ± 6.0 ng m$^{-3}$) and K$^+$ (0.33 ± 0.07 µg m$^{-3}$) with
high OC/EC ratio (7.18 ± 0.2) were observed at the measurement site in Daejeon, Korea.
These results support that the haze episode occurred during the late July, 2014 was
mainly caused by the long-range transport of the smoke plume emitted from Siberia
forest fires. The Siberia smoke episode clearly distinguished compared to the haze
episode caused by long-range transported anthropogenic pollutants emitted from the
East China which showed elevated SO$_4^{2-}$ concentration with weak increases in OC and
levoglucosan concentrations.

Acknowledgement
This work was conducted by a co-research project of National Institute of
Environmental Research (NIER) and Korea Research Institute of Standards and Science
(KRISS). This work was funded by the National Research Foundation under grant NRF-
2015R1C1A1A02036580. We thank to Dr. B. Holben and Dr. M. Panchenko for their
efforts in establishing and maintaining Yakutsk and Ussuriysk AERONET sites in
Russia. The authors gratefully acknowledge the NOAA Air Resources Laboratory (ARL)



for the provision of the HYSPLIT transport and dispersion model and/or READY
website (http://www.arl.noaa.gov/ready.html) used in this publication. The authors also
thank the NASA-US for making available the Collection 005 Level-2 MODIS data.






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



Table 1. Summary of fine particle ($PM_{2.5}$) mass, organic and inorganic chemical composition of $PM_{2.5}$ particles during the Chinese haze and Siberian forest fire episodes measured at Daejeon in Korea during summer, 2014.

| Components | Unit | [1]Chinese Haze | [2]Siberia Forest Fire |
|---|---|---|---|
| | | Range (Average ± 1σ) | |
| $PM_{2.5}$ mass | | 44.5–65.1 (52.3 ± 11.1) | 44.3–56.2 (50.2 ± 8.4) |
| $SO_4^{2-}$ | | 20.9–25.1 (23.1 ± 2.1) | 7.4–9.2 (8.3 ± 1.3) |
| $NO_3^-$ | | 0.9–5.0 (2.8 ± 2.1) | 1.1–1.7 (1.4 ± 0.4) |
| $NH_4^+$ | ($\mu g\ m^{-3}$) | 6.1–12.7 (10.0 ± 3.5) | 4.6–5.4 (5.0 ± 0.6) |
| OC | | 3.6–5.7 (4.8 ± 1.1) | 10.0–11.6 (10.8 ± 1.1) |
| EC | | 1.9–2.2 (2.0 ± 0.2) | 1.4–1.6 (1.5 ± 0.2) |
| $K^+$ | | 0.17–0.33 (0.27 ± 0.08) | 0.28–0.38 (0.33 ± 0.07) |
| OC/EC ratio | | 1.93–2.64 (2.4 ± 0.41) | 7.04–7.32 (7.18 ± 0.19) |
| Levoglucosan | ($ng\ m^{-3}$) | 13.4–35.7 (22.3 ± 11.8) | 115.4–123.9 (119.6 ± 6.0) |
| Mannosan | | 3.0–6.8 (4.5 ± 2.0) | 32.9–37.0 (34.9 ± 2.9) |

[1]Chinese haze: during 14–16 July 2014

[2]Siberia forest fire: during 27–28 July 2014





Table 2. Summary of ratios among biomass burning tracers during the Chinese haze and Siberian forest fire episodes measured at Daejeon in Korea during summer, 2014.

| Components | Chinese Haze | Siberia Forest Fire |
|---|---|---|
| | Range (Average ± 1σ) | |
| Levoglucosan/Mannosan ratio | 4.41–5.22 (4.81 ± 0.41) | 3.35–3.51 (3.43 ± 0.11) |
| Levoglucosan/$K^+$ ratio | 0.05–0.11 (0.08 ± 0.03) | 0.33–0.41 (0.37 ± 0.06) |


Figure captions

Fig. 1. Area map of the measurement site (36.19 °N, 127.24 °E) in Daejeon, Korea
(@Google Map). Siberia forest is located at ~3,000 km north of the Korean
Peninsula.

Fig. 2. Temporal variation of chemical components of fine particulate matter ($PM_{2.5}$) at
the Daejeon site during July, 2014. Daily average $PM_{2.5}$ mass concentrations were
obtained from a beta-attenuation technique.

Fig. 3. (a) MODIS RGB image on 25 July 2014 and (b) air mass backward trajectories
during 26–28 July 2014 when smoke plume originated from the Siberia forest fires
had an impact on the Korean Peninsula. Red, blue, and green in (b) represent air
mass backward trajectories arriving at 200 m, 500 m, and 1000 m heights,
respectively. The Yakutsk site (61.66 °N, 129.37 °E) and Ussuriysk site (43.70 °N,
132.16 °E) in (b) are AERONET sites in Russia. MODIS RGB image in (a) was
obtained        from        the        NASA        Worldview        website
(https://earthdata.nasa.gov/labs/worldview/).

Fig. 4. MODIS aerosol optical depth (AOD) over the East Asia from 23 July to 28 July
2014.

Fig. 5. Temporal variations of AOD measured by a sunphotometer at the Yakutsk site
and Ussuriysk site, Russia during July 2014.

Fig. 6. MODIS RGB images and vertical profiles of total attenuated backscatter at 532
nm measured by the CALIPSO satellite during (a) 24, (b) 25, and (c) 27 July 2014.
Yellow line over MODIS RGB image represents the observation routes of the
CALIPSO satellite which is consistent with x-axis in vertical profiles of total



attenuated backscatter images.

Fig. 7. (a) MODIS RGB image during 14 July 2014 and (b) air mass backward trajectories during 15–16 July 2014 when haze plume originated from the East China had an impact on the Korean Peninsula.

Fig. 8. MODIS AOD over the East Asia during 13–15 July 2014.

Fig. 9. Temporal variations of $PM_{2.5}$ mass, $K^+$, levoglucosan, OC, EC and $SO_4^{2-}$ concentrations at the Daejeon site during the entire measurement period.

Fig. 10. Average $PM_{2.5}$ mass closures during the long-range transported (a) Chinese haze and (b) Siberia forest fires episodes.

Fig. 11. Scatter plots of OC versus (a) levoglucosan and (b) $K^+$ as well as levoglucosan versus (c) $K^+$ and (d) mannosan during the entire measurement period. Filled black and red diamonds represent the Chinese haze and Siberia forest fire episodes, respectively.

Fig. 12. (a) Levoglucosan to mannosan ratios and (b) levoglucosan to $K^+$ ratios obtained from previous chamber studies, extreme smoke episode in Moscow, Russia during summer, 2010, and the Siberia forest fire episode. **Hardwoods**: Fine et al. (2001, 2002, 2004a, 2004b), Schauer et al. (2001), Engling et al. (2006), Schmidl et al. (2008a), Goncalves et al. (2010), Bari et al. (2009); **Softwoods**: Fine et al. (2001, 2002, 2004a, 2004b), Schauer et al. (2001), Engling et al. (2006), Hay et al. (2002), Schmidl et al. (2008a), Goncalves et al. (2010), Iinuma et al. (2007), Cheng et al. (2013); **Grass**: Sullivan et al. (2008); **Crop residue**: Sullivan et al., (2008), Oanh et al. (2011), Sheesley et al. (2003), Engling et al. (2009), Cheng et al. (2013); **Leaf**: Schmidl et al. (2008b); **Moscow smoke:** Popovicheva et al. (2014); **LRT Siberia FF**: This study.







Figure 1

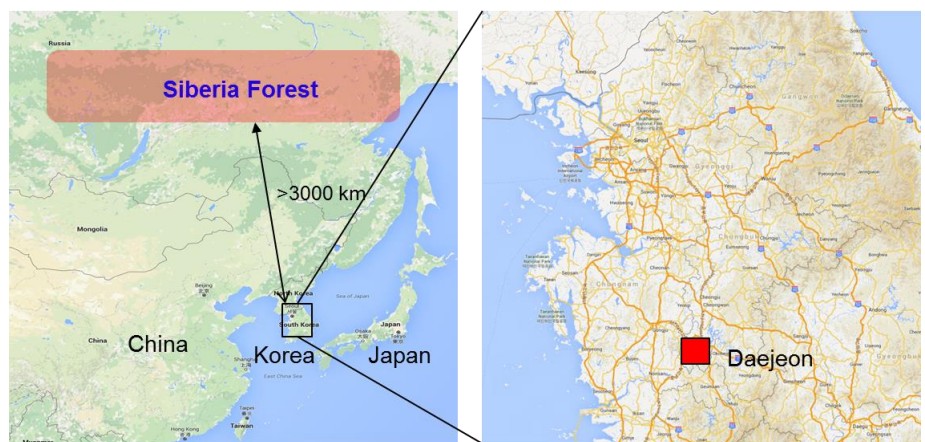





Figure 2

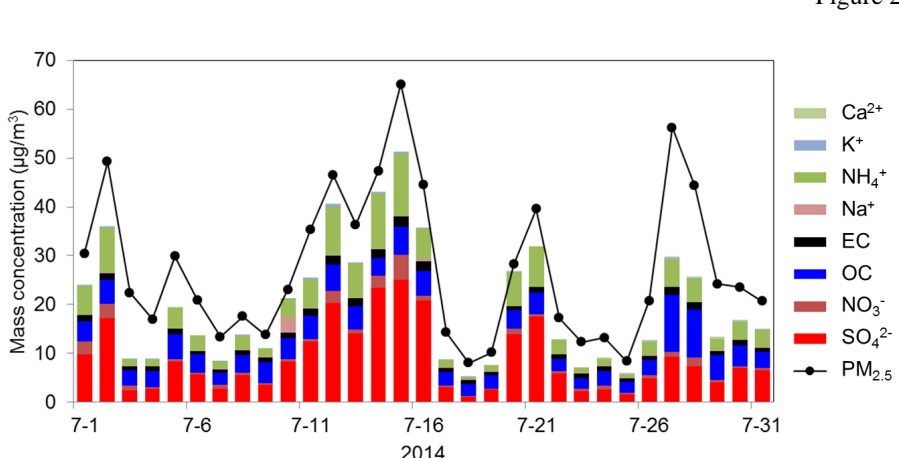





Figure 3

(a) (b)

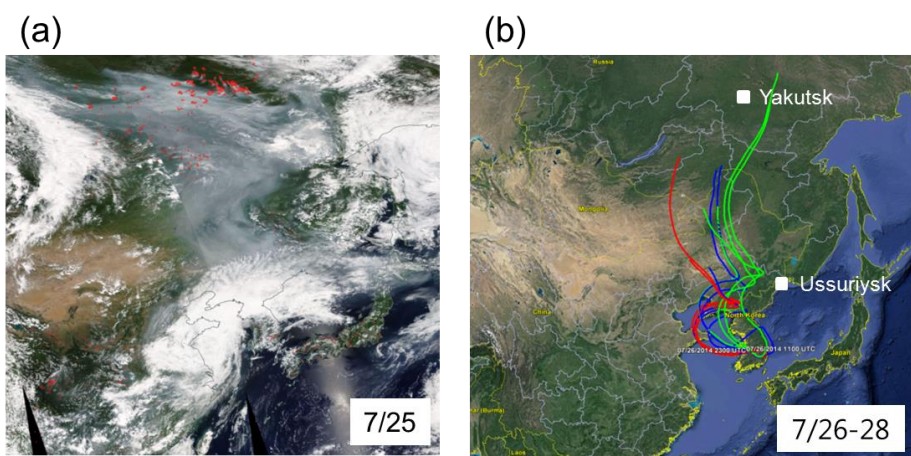

7/25    7/26-28



Figure 4

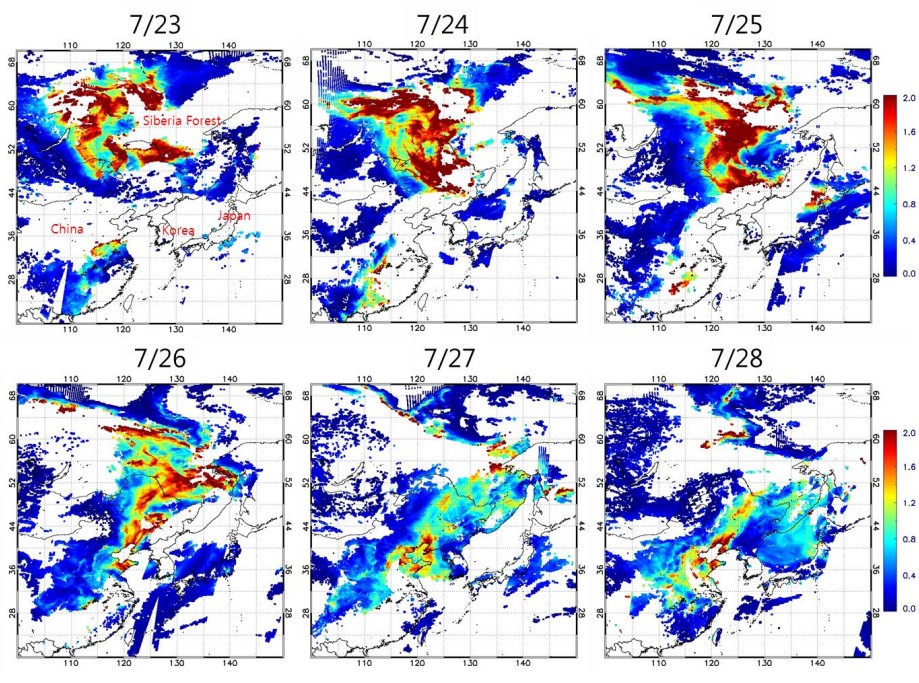





Figure 5

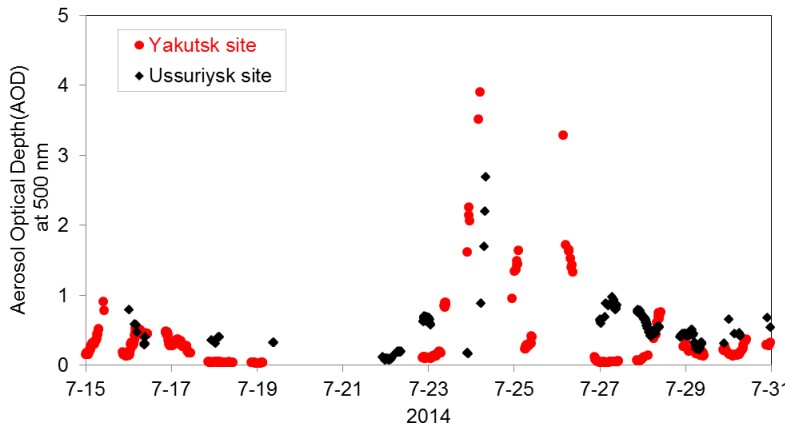



Figure 6

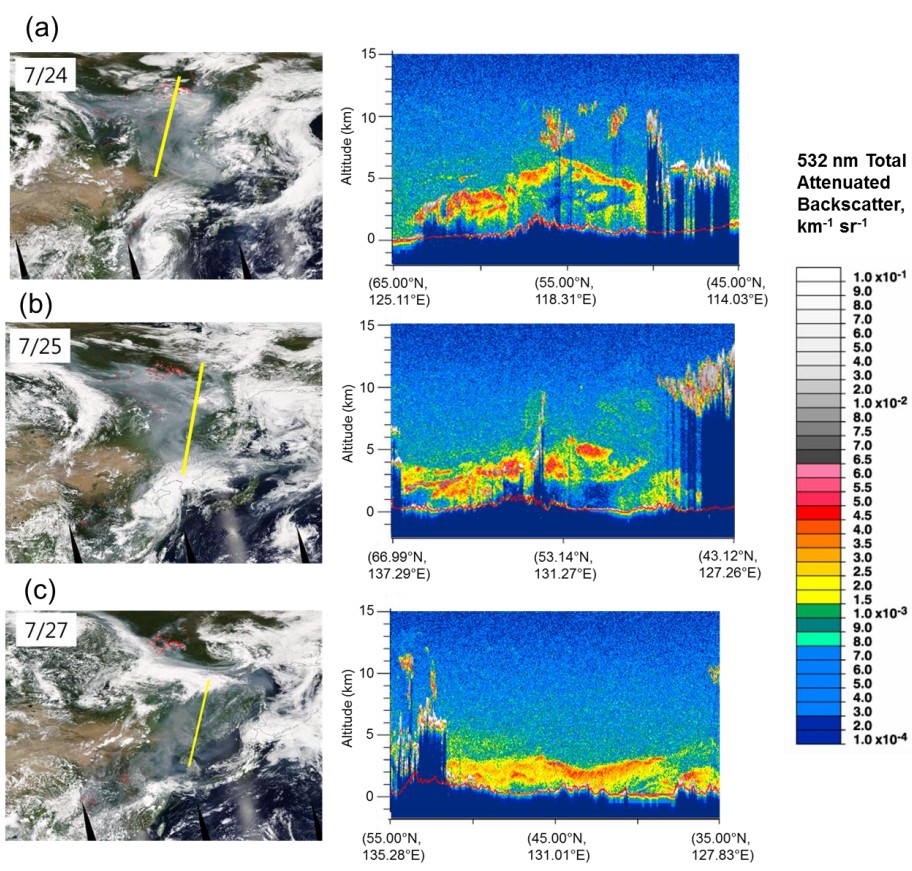



Figure 7

(a)  (b)

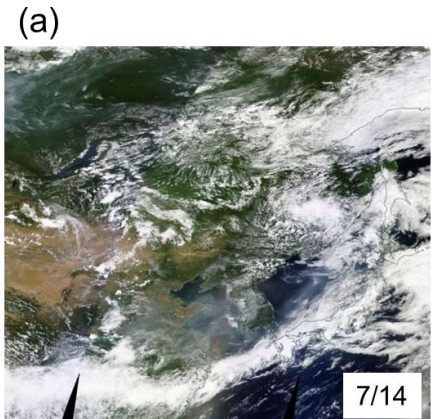
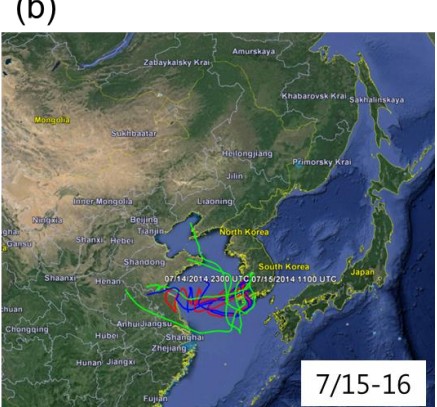



Figure 8

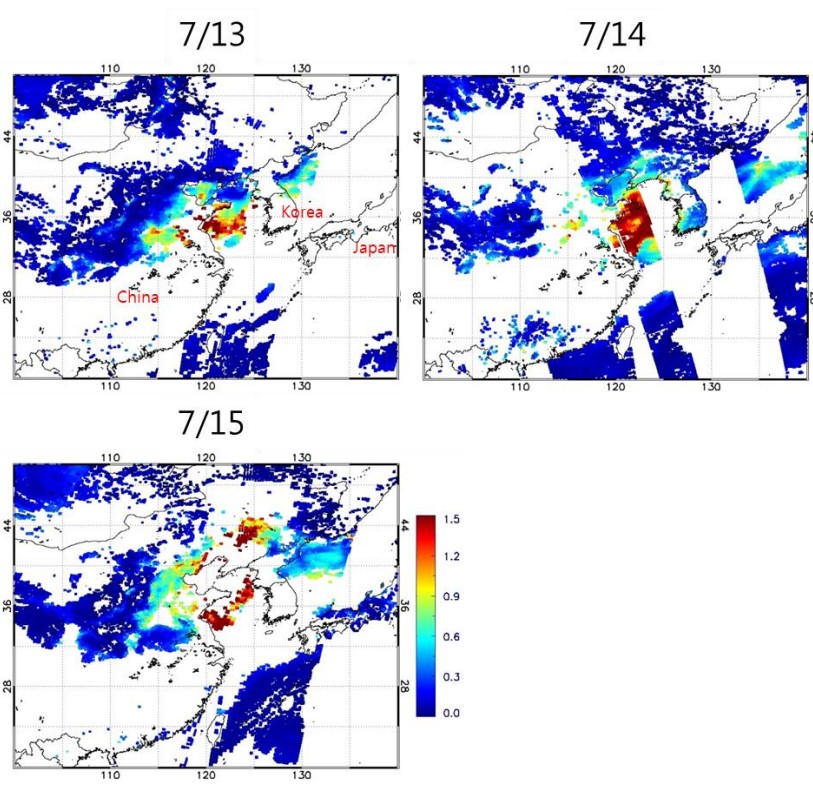





Figure 9

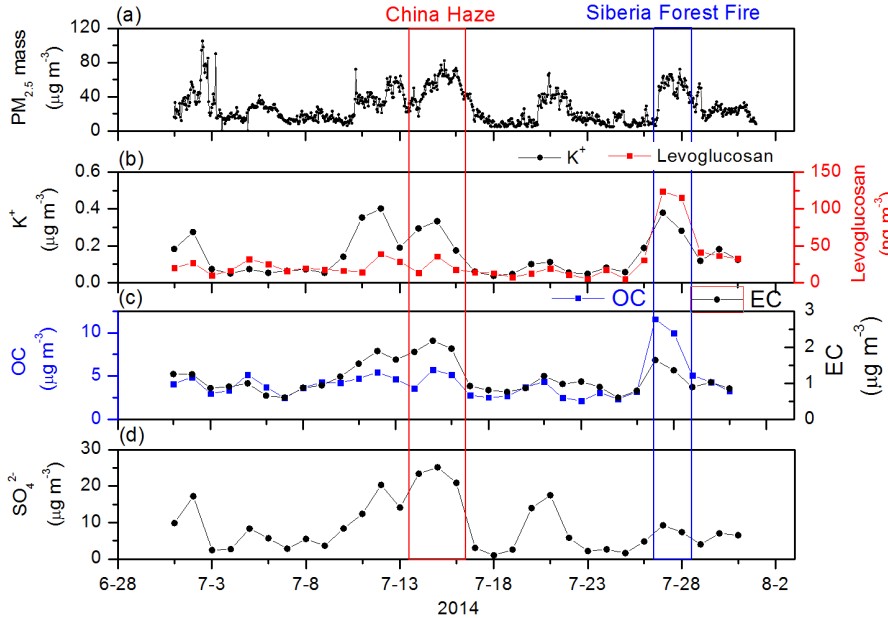



Figure 10

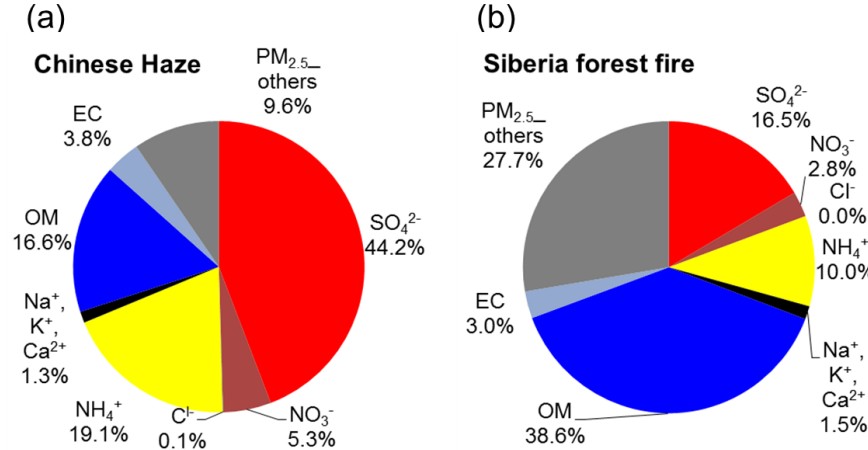



Figure 11

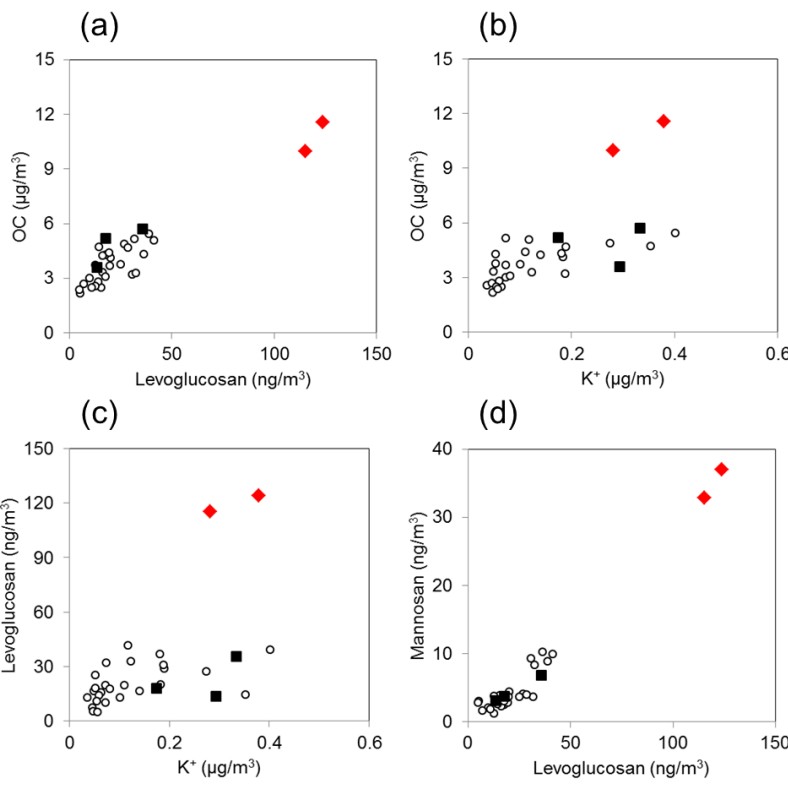



Figure 12

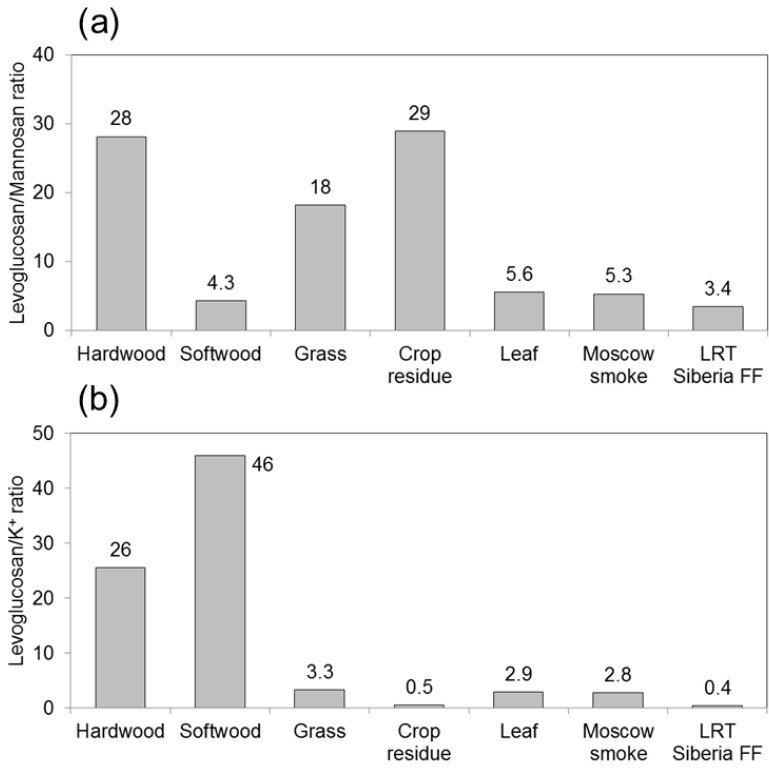