# Peer review of "Impact of Siberian forest fires on the atmosphere over the Korean Peninsula during summer 2014"

_Atmospheric Chemistry and Physics, 2015_

## Referee Comment (RC1) · Anonymous Referee #2 · 15 Feb 2016

General comments:

This manuscript classifies two haze episodes in Korean Peninsula based on different sources, one is from the Siberia forest fire during the late July, 2014, and the other one is from urban and industrial complexes in the East China during the mid July. It also characterizes the chemical compositions of the pollutants during these two haze episodes. This manuscript is well organized, however the presentation of the results part should be improved. You should describe the figure first before you use the information of the figure to support your conclusion.

I have some concerns about the scatter plots in Figure 11. First, I don't know what black circles represent for. I couldn't find description of the black circles anywhere. Second,

the authors mentioned "positive correlation", "poor correlation", or "good correlation" many times, however, it is not convincing to find correlation from only two or three samples. Here, more samples are needed to draw the conclusion on the correlation. Thus, this analysis is not a good support to his conclusion. Third, if you want to show the trends between different chemical compositions, the scatter plot is still not a good tool. Without the time and location of each sample, and so few samples, how could you know the trend is increasing or decreasing with time?

This manuscript actually covers two haze episodes in every analysis and show chemical composition impacts in both haze episodes. Why does the title only include the part of Siberia forest fires?

Overall, I suggest publishing this manuscript after revision based on the comments above and below.

Specific comments:

Line 57. Define PM10.

Line 96. You have to mention that the anthropogenic pollution episode is not in the same period as smoke plumes pollution episode.

Section 2. There are a lot of observations from different sites, and those observations are used in different analyses of this study. I couldn't remember where they come from when I read the later results. I suggest making a table to describe the observation data, include information like where do they come from, site numbers, collecting method, sample frequency, used in which analysis or which figure, etc.

Line 202. The authors only mention two peaks and ignore the peak on 2 July. If you don't want readers to focus on the first peak, you can show the period from 8 July to 31 July. At the beginning of the results section, it is weird to only mention the point that authors want to focus on without explanation of the whole picture. You also need a leading sentence at the beginning of section 3 or at the end of section 3.1 to inform

that you will focus on the "first" and "second" episodes and you are going to show this and that, since you have a very long result section.

Line 220-222. Where did you initialize the HYSPLIT backward trajectories? Did you randomly choose one location in Korean Peninsula or a site location? This information is not mentioned here or in section 2. It is the same issue for Fig 7. The trajectories may pass some parts of the forest, but it is not obviously to see the trajectories pass the red dots from Fig 3a. Maybe there are some red dots covered by the cloud that I couldn't see. The map is not very clear.

Line 234. ADO has dropped to less than 0.5 at late 25 July (Fig 5), and then it increases again. Can you explain this?

Line 237-239. The authors demonstrate that the smoke plumes from Siberia fire would impact Korean peninsula on 27 July and 28 July in the whole manuscript. However, here the authors said the results implied one-day transport. I'm not sure which one is the real conclusion. Moreover, the author concluded that the sharp increase in Ussuriysk site in 24 July was due to the Siberia forest fire without showing any evidence. Is it possible that this sharp increase is due to other sources?

Line 240-249. Poor description. First, describe left column, and then describe right column. Does the right column only represent the Total Attenuated Backscatter along the yellow lines? How did the authors define the paths of yellow lines? All these information should be included in the description.

Line 329-335. Are there only 3 points for Chinese haze episode and 2 points for Siberia forest fire episode? There are too few samples to get any meaningful correlation.

Line 336. "Good correlations". Add the values of the correlations. Please be quantitative.

Line 338. "different correlation patterns". I didn't see obvious difference from the figure. Could you describe more clearly about the patterns' difference?

[Figure]

Line 349. "Poor correlations". Please be quantitative.

Figure 12. I suggest changing the color of the last bar in order to distinguish this study from other referenced studies.
[Figure]

---

## Referee Comment (RC2) · Anonymous Referee #1 · 27 Mar 2016

Dear Authors, Thank you for this manuscript. It describes an interesting case study of long-range transport of Siberian smoke to Korea. I think this paper will produce an interesting contribution to ACP. My main question regarding your analysis is that you do not discuss the potential contribution of biofuel to the southern China haze event. Many literature sources indicate that biofuel is a significant contributor to the energy mix and to the air pollution in rural Chinese areas. I think your analysis would be strengthened if you examined the chemical composition of the southern Chinese haze in the context of literature estimates of biofuel consumption in the southern Chinese region. The recent paper by Rongrong Wu et al. (doi:10.1016/j.atmosenv.2015.12.015) would be a good place to start. Apart from that, this paper is scientifically sound and the conclusions

are reasonable. I believe this paper would benefit from a thorough editing to improve grammar and remove typographical errors. Best of luck with your revisions, and thank you again.
* * *

---

## Author Comment (AC1) · 6 May 2016

**Manuscript #: acp-2015-1022**

**Title: Impact of Siberia forest fires on the atmospheric environment over the Korean Peninsula during summer 2014**

Authors: Jinsang Jung et al.

**Responses to the reviewer's specific comments and questions;**

**Reviewer #1 (Comments):**

**General comments:**

This manuscript classifies two haze episodes in Korean Peninsula based on different sources, one is from the Siberia forest fire during the late July, 2014, and the other one is from urban and industrial complexes in the East China during the mid July. It also characterizes the chemical compositions of the pollutants during these two haze episodes. This manuscript is well organized, however the presentation of the results part should be improved. You should describe the figure first before you use the information of the figure to support your conclusion.

**Specific comments:**
*I have some concerns about the scatter plots in Figure 11. First, I don't know what black circles represent for. I couldn't find description of the black circles anywhere.*
**Response:** Following sentence has been added in the caption of Fig. 11 in the revised MS.
"Open black circles represent the remaining sampling days in July 2014."

*Second, the authors mentioned "positive correlation", "poor correlation", or "good correlation" many times, however, it is not convincing to find correlation from only two or three samples. Here, more samples are needed to draw the conclusion on the correlation. Thus, this analysis is not a good support to his conclusion.*
**Response:** We agreed to the reviewer' comment. We decided to remove the terms "positive correlation", "poor correlation", or "good correlation" in the revised MS. Specific changes can be seen in the late part of this revision document.

*Third, if you want to show the trends between different chemical compositions, the scatter plot*

*is still not a good tool. Without the time and location of each sample, and so few samples, how could you know the trend is increasing or decreasing with time?*

**Response:** We agreed to the reviewer' comment. We decided to remove the terms "positive correlation", "poor correlation", or "good correlation" in the revised MS.

*This manuscript actually covers two haze episodes in every analysis and show chemical composition impacts in both haze episodes. Why does the title only include the part of Siberia forest fires?*

**Response:** Thank you for the comment. Long-range transport of the Siberia forest fire to the Korean Peninsula rarely happen throughout season. However, long-range transport of the Chinese haze are frequently observed and studied. Thus, we want to focus more on the impact of the Siberia forest fire in the title of the manuscript.

*Line 57. Define PM10.*

**Response:** The phrase "(particulate matter with a diameter of $\leq 10$ µm)" has been added in line 56 in the revised MS.

*Line 96. You have to mention that the anthropogenic pollution episode is not in the same period as smoke plumes pollution episode.*

**Response:** The phrase "in the middle of July 2014" has been added in lines 96-97 in the revised MS.

*Section 2. There are a lot of observations from different sites, and those observations are used in different analyses of this study. I couldn't remember where they come from when I read the later results. I suggest making a table to describe the observation data, include information like where do they come from, site numbers, collecting method, sample frequency, used in which analysis or which figure, etc.*

**Response:** Thank you for the suggestion. We decided to add a table containing summary of measurement parameters and conditions. Following sentence has been added in line 102 in the revised MS.

"Table 1 summaries the measurement parameters and conditions of this study."

Following table has been added in table 1 in the revised MS.

Table 1. Measurement parameters and conditions of this study.

| Measurement parameters | Site | Sampling method | Measurement method | Data frequency |
|---|---|---|---|---|
| $PM_{2.5}$ mass | Daejeon, | Online | Beta-attenuation | 1 h |

| | Korea | measurement | monitor | |
|---|---|---|---|---|
| Levoglucosan, Mannosan | Daejeon, Korea | $PM_{2.5}$ filter sampling | High-performance anion-exchange chromatography | 1 day |
| Water-soluble ions ($NO_3^-$, $SO_4^{2-}$, etc) | Daejeon, Korea | $PM_{2.5}$ filter sampling | Ion Chromatography | 1 day |
| Organic carbon (OC), elemental carbon (EC) | Daejeon, Korea | Online measurement | Semi-continuous OC/EC analyzer | 1 h |
| Aerosol optical depth (AOD) | Yakutsk and Ussuriysk, Russia | Online measurement | Sunphotometer | ~15 min |

*Line 202. The authors only mention two peaks and ignore the peak on 2 July. If you don't want readers to focus on the first peak, you can show the period from 8 July to 31 July. At the beginning of the results section, it is weird to only mention the point that authors want to focus on without explanation of the whole picture. You also need a leading sentence at the beginning of section 3 or at the end of section 3.1 to inform that you will focus on the "first" and "second" episodes and you are going to show this and that, since you have a very long result section.*

**Response:** Thank you for the comment. As reviewer suggested, we have decided to show data from 4 July to 31 July not to confuse readers. The study period of figure 2 and figure 9 were modified as 4 July to 31 July. Please see the modified figure 2 and figure 9 in the revised MS.

[Figure]

Fig. 2. Temporal variations in the chemical components of fine particulate matter

(PM$_{2.5}$) at the Daejeon site during July 2014.

[Figure]

Fig. 9. Temporal variations in PM$_{2.5}$ mass, K$^+$, levoglucosan, OC, EC, and SO$_4^{2-}$ concentrations at the Daejeon site over the entire measurement period.

*Line 220-222. Where did you initialize the HYSPLIT backward trajectories? Did you randomly choose one location in Korean Peninsula or a site location? This information is not mentioned here or in section 2. It is the same issue for Fig 7. The trajectories may pass some parts of the forest, but it is not obviously to see the trajectories pass the red dots from Fig 3a. Maybe there are some red dots covered by the cloud that I couldn't see. The map is not very clear.*

**Response:** Following phrase has been added in lines 176-177 in the revised MS.

"at the sampling site (36.19 °N, 127.24 °E) in Daejeon, Korea"

As we already mentioned in lines 180-183, the HYSPLIT backward trajectory can be used to track general airflow pattern rather than the exact pathway of air masses. As shown in figure 3, we can clearly see similar movement of the Siberia smoke plume from MODIS RGB image in figure 3a compared to the HYSPLIT backward trajectories in figure 3b.

*Line 234. ADO has dropped to less than 0.5 at late 25 July (Fig 5), and then it increases again. Can you explain this?*

**Response:** Following sentences have been added in lines 234-238 in the revised MS.

"The AOD dropped to <0.5 during 6:00−10:00 UTC, 25 July and increased again during 26 July.

Because high AOD at the Yakutsk site was caused by transport of the Siberian smoke plume (Fig. 3), the sharp drop in AOD observed during 25 July can be explained by a change in wind direction at the Yakutsk site."

*Line 237-239. The authors demonstrate that the smoke plumes from Siberia fire would impact Korean peninsula on 27 July and 28 July in the whole manuscript. However, here the authors said the results implied one-day transport. I'm not sure which one is the real conclusion.*

**Response:** We agreed to the reviewer' comment. Sentence in lines 244-245 in the original MS has been modified as follows.

"These results again suggest the transport of Siberian smoke plumes to the northern Korean Peninsula."

*Line 237-239. Moreover, the author concluded that the sharp increase in Ussuriysk site in 24 July was due to the Siberia forest fire without showing any evidence. Is it possible that this sharp increase is due to other sources?*

**Response:** Following sentence has been added in lines 242-243 in the revised MS.

"Spatial distributions of AOD from the MODIS satellite data (Fig. 4) clearly show that the Siberian smoke plumes extended over the Ussuriysk site during 24 July 2014."

*Line 240-249. Poor description. First, describe left column, and then describe right column. Does the right column only represent the Total Attenuated Backscatter along the yellow lines? How did the authors define the paths of yellow lines? All these information should be included in the description.*

**Response:** A paragraph in lines 246-258 in the original MS has been revised as follows.

"Figure 6 shows MODIS RGB images and vertical distributions of total attenuated backscatter at a wavelength of 532 nm measured by the CALIPSO satellite during 24, 25, and 27 July 2014. The left column in Fig. 6 shows MODIS RGB images taken during the Siberian smoke episode. These images show smoke plumes originating from the Siberian forest and being transported over northeastern China. The yellow lines over the images in the left column of Fig. 6 indicate the route of the CALIPSO satellite, and correspond to the x-axis of the backscatter plots shown in the right column of Fig. 6. In the total attenuated backscatter measurement plots (Fig. 6, right), red and yellow represent atmospheric aerosol particles and white represents clouds. Figure 6a and b clearly show that between 24 and 25 July 2014, a smoke layer existed approximately 3–5 km in height near the source region of the Siberian forest fires. As shown in Fig. 6c, the height of the smoke layer decreased to below 2 km on 27 July 2014 as it reached the Korean Peninsula."

*Line 329-335. Are there only 3 points for Chinese haze episode and 2 points for Siberia forest fire episode? There are too few samples to get any meaningful correlation.*

**Response:** We agreed with the reviewer' comment. Following sentence was deleted.

"Positive correlation was obtained between levoglucosan and OC concentrations during the Siberia forest fire and Chinese haze episodes in Fig. 11a."

Lines in 329-333 in the original MS have been modified as follows. Please see lines 352-357 in the revised MS.

"OC concentrations increased as levoglucosan and $K^+$ concentrations increased during the Siberian forest fire episode (Fig. 11a). Elevated OC/EC ratios were also observed during the Siberian forest fire episode (7.18 ± 0.2). Simultaneous increases in $K^+$, OC (Fig. 11b), and levoglucosan concentrations (Fig. 11c) during the Siberian forest fire episode suggest that the $K^+$ originated primarily from the smoke plume during the Siberian forest fire episode."

*Line 336. "Good correlations". Add the values of the correlations. Please be quantitative.*
*Line 338. "different correlation patterns". I didn't see obvious difference from the figure. Could you describe more clearly about the patterns' difference?*

**Response:** Thank you for the comment. After considering reviewer's comments in lines 336 and 338, the sentences starting "Good correlations of $K^+$ …" in lines 336-340 have been revised as follows. Please see lines 358-364 in the revised MS.

"OC and levoglucosan concentrations observed during the Chinese haze episode are similar to those observed during the non-episode period, as shown in Fig. 11a. However, small increases in $K^+$ concentration were observed during the Chinese haze episode, as shown in Fig. 11b, resulting in relatively small levoglucosan/K+ ratios during the Chinese haze episode (0.08 ± 0.03) compared with those during the Siberian forest fire episode (0.37 ± 0.06). This difference in levoglucosan/$K^+$ ratios can be explained as follows."

*Line 349. "Poor correlations". Please be quantitative.*

**Response:** The phrase "Poor correlations of $K^+$ with OC and levoglucosan concentrations during the Chinese haze episode suggest" in lines 373-375 in the original MS has been modified as follows.

"The lack of significant increases in OC/EC ratio (2.4 ± 0.4), and OC and levoglucosan concentrations during the Chinese haze episode compared with non-episode measurements suggests"

*Figure 12. I suggest changing the color of the last bar in order to distinguish this study from*

*other referenced studies.*

**Response:** Thank you for the comment. The last bar of Fig. 12 was changed as follows.

[Figure]

---

## Author Comment (AC2) · 6 May 2016

**Manuscript #: acp-2015-1022**

**Title: Impact of Siberia forest fires on the atmospheric environment over the Korean Peninsula during summer 2014**

Authors: Jinsang Jung et al.

**Responses to the reviewer's specific comments and questions;**

**Reviewer #2 (Comments):**

**General Comments:**

Dear Authors, Thank you for this manuscript. It describes an interesting case study of long-range transport of Siberian smoke to Korea. I think this paper will produce an interesting contribution to ACP.

My main question regarding your analysis is that you do not discuss the potential contribution of biofuel to the southern China haze event. Many literature sources indicate that biofuel is a significant contributor to the energy mix and to the air pollution in rural Chinese areas. I think your analysis would be strengthened if you examined the chemical composition of the southern Chinese haze in the context of literature estimates of biofuel consumption in the southern Chinese region. The recent paper by Rongrong Wu et al. (doi:10.1016/j.atmosenv.2015.12.015) would be a good place to start.

**Response:** Thank you for the comment. Following paragraph has been added in lines 298-314 in the revised MS.

"It has been reported that biomass burning (including biofuel) contributed 14.1% of the total VOC emissions in China during 2012, whereas in Anhui province the contribution of biomass combustion to VOC emissions was 28.7% (Wu et al., 2016). Li et al. (2015) reported that biomass burning contributed 58% of OC in Nanjing, China during summer 2012, suggesting that biomass burning is the dominant source of OC in this region. Du et al. (2011) classified the haze events in Shanghai, China during summer 2009 into three categories: biomass-burning induced (high $K^+$, low $SO_4^{2-}$ and $NO_3^-$), complicated (high $SO_4^{2-}$ and $NO_3^-$, good correlation between $K^+$ and $SO_4^{2-}$ and $NO_3^-$), and secondary (high $SO_4^{2-}$ and $NO_3^-$, low $K^+$) pollution. Because Anhui, Nanjing, and Shanghai are located near the source of the long-range transported

Chinese haze (Fig. 8), the chemical composition of pollution in those areas can be used to understand the Chinese haze episode observed in this study. Temporal patterns in $K^+$ concentration are similar to those of $SO_4^{2-}$, and a sharp increase in $SO_4^{2-}$ concentration was observed during the Chinese haze episode (Fig. 9). This type of pollution episode is similar to the 'complicated' pollution described by Du et al. (2011), and suggests that the Chinese haze episode was caused mainly by secondary aerosol such as $SO_4^{2-}$ and $NH_4^+$, rather than by biomass burning emissions."

Three references were added in the reference section.
Du, H., Kong, L, Cheng, T., Chen, J., Du, J., Li, L., Xia, X., Leng, C., and Huang, G.: Insights into summertime haze pollution events over Shanghai based on online water-soluble ionic composition of aerosols, Atmos. Environ., 45, 5131−5137, 2011.
Li, B., Zhang, J., Zhao, Y., Yuan, S., Zhao, Q., Shen, G., and Wu, H.: Seasonal variation of urban carbonaceous aerosols in a typical city Nanjing in Yangtze River Delta, China, Atmos. Environ., 106, 223−231, 2015.
Wu, R., Bo, Y., Li, J., Li, L., Li, Y., and Xie, S.: Method to establish the emission inventory of anthropogenic volatile organic compounds in China and its application in the period 2008-2012, Atmos. Environ., 127, 244−254, 2016.

Apart from that, this paper is scientifically sound and the conclusions are reasonable. I believe this paper would benefit from a thorough editing to improve grammar and remove typographical errors. Best of luck with your revisions, and thank you again.
**Response:** Thank you for the comment. The revised MS has been proofread by a English native speaker.